# Fear of Losing Jobs during COVID-19: Can Psychological Capital Alleviate Job Insecurity and Job Stress?

**DOI:** 10.3390/bs12060168

**Published:** 2022-05-30

**Authors:** Bangxin Peng, Wisanupong Potipiroon

**Affiliations:** Faculty of Management Sciences, Prince of Songkla University, Hatyai 90112, Thailand; wisanupong.p@psu.ac.th

**Keywords:** job insecurity, psychological capital, job stress, bank tellers, COVID-19, multilevel structural equation modeling (MSEM)

## Abstract

Job insecurity is a growing concern among bank employees. In this research, we examined whether psychological capital can alleviate bank workers’ perceptions of job insecurity and job stress during COVID-19. In particular, we aimed to examine this relationship at both the individual and work-unit levels of analysis. Based on the data collected from 520 bank tellers in 53 bank branches in Thailand, our multilevel structural equation modeling (MSEM) analyses revealed that there was significant between-branch variation in the levels of psychological capital (15%), job insecurity (23%) and job stress (24%). In particular, perceptions of job insecurity were found to have a positive effect on job stress at both levels of analysis. We also found that psychological capital significantly reduced perceptions of job insecurity and job stress at the individual level. These findings emphasize that perceptions of job insecurity can emerge at both the individual and work-unit levels. Theoretical and practical implications are discussed.

## 1. Introduction

Traditionally, bank tellers play a particularly important role in delivering high quality in-person services to customers. However, the advent of new technology (e.g., fintech) has ushered in automated banking services that are now replacing the traditional counter services [1]. The shift to digital banking has led several major banks around the globe to radically transform and slash their workforces. For example, in Thailand, it has been reported that about 800 bank branches of the six largest banks were closed off in 2022 [2]. The COVID-19 pandemic has further exacerbated this situation as it has expedited the pace of automation in retail banking [1,3]. This trend might be irreversible as consumers are now preferring automated services over in-person services [1,4]. As a result, bank workers around the world are experiencing growing concerns about their future employment [1].

In this research, we aim to examine the impact of bank tellers’ perceived job insecurity on their job stress amidst the COVID-19 pandemic. Job insecurity is generally defined as “an overall concern about the continued existence of the job in the future” [5]. Past research has shown that when employees experience job insecurity, they tend to experience a decline in sleep quality [6], organizational commitment [7], productivity [8], and willingness to innovate [9], and also an increase in blood pressure and turnover intentions [10].

Job stress could be another important negative outcome of perceived job insecurity [11,12,13,14,15]. Job stress can be viewed as a person’s feelings of anxiety and worry that occur as a result of perceived threats and undesirable conditions in the work environment, which are physically and psychologically difficult to cope with [16,17,18,19]. Workers in various sectors, e.g., those on the front line [20], hospitality workers [21], and bank employees [22] are now experiencing increasing levels of job stress due to COVID-19. When individuals experience stress at work, it has been shown to undermine their job performance [22,23,24], work presenteeism [25], sleep quality [26], and health outcomes [27].

This research draws from the Transactional Theory of Stress and Coping [28] to propose that psychological capital is an important personal resource that can be used to cope with perceptions of job security. In particular, psychological capital has been defined as a personal positive psychological state characterized by hope, efficacy, resilience, and optimism [29]. Past research indicates that psychological capital is an important personal resource for dealing with perceived job insecurity [30,31,32,33] A recent meta-analysis on job insecurity also indicates that, in comparison to job demands, job resources, such as psychological capital, may have a stronger impact on job insecurity perceptions [34].

Our study aims to extend past research by adopting a multilevel perspective to examine the relationship between psychological capital, perceived job insecurity, and job stress at both the individual and work-unit levels of analysis. In the context of this research, the work-unit level of analysis refers to the branches of banks in which we believe there could be significant variation in employees’ perceptions. Past research indicates that individuals within the same work units may share similar levels of psychological capital (i.e., collective psychological capital) [35], job insecurity (i.e., job insecurity climate) [6,36,37,38], and job stress (i.e., collective stress) [14,39]. To illustrate this point, individuals within the same work units may be affected by the same directives from the top management and human resources (HR) policies, such that there could be shared perceptions about their future employment and job stress levels that are associated with specific organizational changes. The knowledge from this multilevel approach could inform us about the different strategies for dealing with perceptions of job insecurity and job stress. This study contributes to the behavioral science and management literatures by shedding light on important psychological resources that individuals and organizations can use to mitigate the undesirable effects of job insecurity perceptions. Towards the end of this paper, we discuss what organizations can do to enhance employees’ psychological capital. In the sections below, we discuss the relationship between job insecurity and job stress and how psychological capital can help to mitigate these negative perceptions.

## 2. Theory and Hypotheses

### 2.1. Transactional Theory of Stress and Coping

Job stress has been defined differently in the literature but most definitions view it as a person’s emotional response to the undesirable stimuli in the work environment. For example, job stress has been described as a person’s “feeling of personal dysfunction as a result of perceived conditions or happenings in the work setting”. Jamal [16] also defined job stress as an “individual’s reactions to the characteristics of work environment which appear threatening”. In particular, job stress occurs “when the demands of the environment exceed (or threaten to exceed) a person’s capabilities and resources to meet them or the needs of the person are not being supplied by the job environment” [19]. Similarly, Lazarus [17] described stress as feelings of anxiety and worry that are physically and psychologically difficult to confront.

Relying on these definitions, we view job stress as a person’s negative feelings that occur as a reaction to threatening or undesirable conditions in the work environment that go beyond one’s coping ability. In so doing, we draw from the Transactional Theory of Stress and Coping [28] as a theoretical lens for the investigating the negative impact of job insecurity perceptions on job stress and how individuals’ psychological capital can help alleviate the adverse impact of job insecurity perceptions. This theory posits that individuals constantly appraise stimuli within their work environment, which, in turn, trigger specific emotions and feelings. When such stimuli are perceived to be threatening to the self, individuals will try to engage in coping strategies to manage negative emotions and feelings [28].

Two types of cognitive appraisals are involved in this process. Primary appraisal ascribes meaning to a specific individual-environmental transaction and determines the significance of that transaction to an individual’s well-being (e.g., how does this event affect me?) [28]. In the workplace, stimuli that are appraised as threatening are also referred to as work-related stressors, which can be categorized as “eventful” stressors and “chronic” stressors [40]. Specifically, eventful stressors refer to unexpected events that occur at specific points in time, whereas chronic stressors occur repeatedly over a considerable period. For example, an unexpected layoff could be viewed as an eventful stressor, whereas a constant fear of losing jobs (i.e., perceptions of job insecurity) during COVID-19 could be characterized as a chronic stressor.

In contrast to primary appraisal, secondary appraisal determines what can be done to manage a stressful situation [28]. The result of this appraisal process is known as “coping”—a self-defense mechanism in dealing with threats to one’s psychological integrity [17] For example, individuals may actively seek to avoid the stressors or to minimize the harmful effects of the stressors by engaging in various coping strategies such as positive appraisal or planful problem solving [28]. Past research indicates that personality traits and individual predispositions can serve as personal resources in helping individuals resist the deleterious effects of stress. For example, positive affectivity [41], internal locus of control [42], and perceptions of workplace control [10] have been shown to reduce the adverse impact of perceived job insecurity. As discussed further below, we draw attention to the role of psychological capital in mitigating the negative effects of perceived job insecurity and job stress [34,43].

### 2.2. Perceptions of Job Insecurity and Job Stress

Job insecurity has been defined as “a sense of powerlessness” that results from the potential perceived threat to the continuity of one’s job [5,44]. While there is no consensus on the measurement of the job insecurity concept [45,46], with some studies measuring it as a multidimensional measure [47] and others measuring it as a unidimensional measure [48], there is a general consensus that job insecurity is an individual’s subjective perceptions as individuals can differ significantly in their perceived job insecurity even though they are exposed to the same stressful event in their employment [13]. Indeed, De Witte [49] argued that perceptions of job insecurity may have an even more profound impact than job loss itself.

Various factors have been shown to influence perceptions of job insecurity [34]. Quite obviously, perceptions of job insecurity are often a direct result of sudden organizational changes involving layoffs [14] and organizational restructuring [50,51], whereas other factors, such as company performance [41], can alleviate these negative perceptions. It has also been shown that younger workers [7], blue-collar and temporary workers [42], and those with a lower level of technology usage [52] are more prone to experiencing higher job insecurity than other groups of employees.

The introduction of new technology (e.g., automation and artificial intelligence [AI]) also contributes significantly to perceptions of job insecurity among automatable occupations [53]. For example, Erebak and Turgut [54] found that employees’ growing concerns about the speed of robot technology led to perceptions of job insecurity. Koo et al. [55] also showed that hotel employees’ perceptions of job insecurity which were caused by the introduction of AI had a significant negative impact on their job engagement and turnover intentions. Similarly, Lingmont and Alexiou [56] showed that individuals’ awareness that their jobs are being automated by STARA (i.e., smart technology, artificial intelligence robotics and algorithms) can have a negative effect on perceptions of job insecurity especially among those whose jobs are at a relatively high risk of being automated including transportation and logistics, construction, sales, education, maintenance, office support, financial and legal service, and manufacturing and agriculture.

This research aims to examine the extent to which perceived job insecurity leads to an increase in job stress among bank employees during COVID-19. As noted earlier, the restrictive measures that have been adopted by governments worldwide in their attempt to counteract the effects of the pandemic have led to a retrenchment of workforce and significant changes in the work arrangements (e.g., working from home (WFH)) in various sectors [1]. Several studies have been conducted to investigate the impact of job insecurity during the unfolding disaster of COVID-19. For example, Aguiar-Quintana [23] showed that perceived job insecurity during COVID-19 had an adverse impact on the job performance of hotel employees in Spain by increasing their anxiety and depression. In another study of hotel workers in Pakistan, Abbas et al. [57] reported that perceptions of job insecurity that result from the COVID-19 pandemic had a negative effect on employees’ self-esteem and a positive effect on their perceptions of economic deprivation. Moreover, Nemteanu et al. [58] reported that perceptions of job instability during COVID-19 had a significant negative impact on work satisfaction, satisfaction with supervisor support, and perceived promotion opportunities among employees in Romania. Finally, Ganson et al. [59] reported a significant association between job insecurity and symptoms of anxiety and depression among U.S. young adults during COVID-19. Despite these notable findings, research has yet to examine the possible negative impact of job insecurity among the banking workforce. This leads to the first hypothesis.

**Hypothesis** **1.**
*Perceived job insecurity is positively related to job stress at the individual level of analysis.*


In addition to the first hypothesis, it is important to acknowledge that worrying about job loss is not only an individual-level phenomenon (i.e., in terms of personal and private concerns) but it can also occur at a work-unit level (i.e., in terms of shared perceived climate among employees) [36,37,60,61]. In particular, perceived job insecurity may manifest itself through conversations and rumors about the introduction of new technology and future employment within the organization. Such interactions may give rise to the perceived climate of job insecurity, which in turn results in shared job stress at the work-unit level. Indeed, it has been shown that workers in manufacturing plants that were severely downsized experienced significantly higher levels of work stress than do workers at other plants [39]. Låstad et al. [60] also showed that employees in Switzerland who perceived higher levels of job insecurity climate reported higher levels of health symptoms and burnout. Moreover, in a study of human resources (HR) employees in Taiwan, Hsieh et al. [36] found that perceptions of job insecurity climate led to lower levels of work engagement and job satisfaction via the mediating role of perceived organizational obstruction. This leads to the following hypothesis.

**Hypothesis** **2.**
*Perceptions of job insecurity climate are positively related to collective stress at the branch level of analysis.*


### 2.3. The Role of Psychological Capital

We further propose that psychological capital can mitigate perceptions of job insecurity. Psychological capital is defined as an individual’s positive psychological state of development, which is characterized by self-efficacy, optimism, hope, and resilience [29]. While psychological capital is viewed as a manifestation of individuals’ personality traits and strengths, which are relatively stable over time, it is generally viewed as a state-like construct, which is malleable and can be developed through training and interventions [43]. Past research indicates that psychological capital is a constructive personal resource [34,62,63] that allows individuals to appraise an adverse job environment positively, in turn diminishing the perceived severity of job insecurity. We propose that individuals with high psychological capital are likely able to effectively deal with unforeseen situations by developing positive appraisal of threatening events, thereby reducing perceptions of job insecurity. Below, we explain how each of the psychological capital components can allow individuals to do so.

First, individuals with hope are characterized by strong willpower and persistence in goal accomplishment [62], which in turn encourages individuals to search for alternative pathways when the original ways of achieving goals are not available [64]. For example, hopeful employees may consciously set new goals and search for new ways to prepare for re-employment rather than wait for job loss to occur [65].

Secondly, individuals with high self-efficacy tend to thrive on challenges and do not avoid obstacles in their ways [62]. This characteristic allows individuals to build an “I can do it” self-concept [66]. With more self-efficacy, the individual is able to identify and garner cognitive resources and motivation to complete the specific task [62]. Individuals with high levels of self-efficacy are not only able to summon the resources that exist in the organization to accomplish their tasks effectively, but when faced with the probability of job loss, they have the confidence to take proactive actions to ensure their job search properness such as by developing new skills to cope with the threats of job insecurity.

Third, individuals who are resilient are able to engage in positive adjustment and adaption to adversity [67]. Individuals with high resilience are able to withstand and recover from major stress more easily [68], while also experiencing more positive emotions and fewer negative emotions in the process [69]. This scenario can be attributed to the ability to use minimal attention or cognitive efforts as resilient attitudes allow individuals to cope with stressful situations automatically [70]. Individuals with high resilience are also able to “accept the source of stress” and “learn new way to adapt” [70,71]. Furthermore, individuals with high resilience have the vigor to avert from adversity and return to normal state; in such a situation, they “thrive” rather than “survive” [72]. Thus, resilient workers can withstand the perceived difficulties associated with future loss of employment and respond to the challenges by trying to understand the causes of perceived job insecurity while also exploring new pathways to increase their job security [23].

Fourth, individuals with high optimism can translate negative events as external, temporary, and situational factors [73,74]. One important resource that optimists have is the ability to see the future in favorable terms instead of being depressed and taken aback by the obstacles they encounter [62]. Optimistic individuals are less likely to cope with job demand by withdrawing [75]. Instead, they develop skills to be able to cope with challenges and make full use of the opportunity that exists in the environment, which is facilitated by the willful assessment of the lessons learned [76]. Such positive expectations for future success thus encourage optimistic individuals to adapt themselves to the challenges of job insecurity by working even harder or further self-development [12].

Individuals’ actions that emanate from these specific components of psychological capital are similar to the coping strategies proposed by Folkman and Lazarus [77]. In particular, a person with high psychological capital may engage in different kinds of coping strategies when faced with the prospect of losing jobs, for example, by engaging in confrontive coping (e.g., I will fight for what I want), distancing (e.g., I will make light of the situation and refuse to get too serious about it), seeking social support (e.g., I will talk to someone who could do something concrete about the problem), planful problem solving (e.g., I will make a plan of action and followed it), and positive reappraisal (e.g., I may come out of the experience better than when I went in).

In this respect, previous studies have shown that psychological capital plays an important role in mitigating perceptions of job insecurity. For example, in a study of hotel employees in Iran, Darvishmotevali and Ali [12] showed that psychological capital buffered the negative impact of job insecurity on employees’ subjective well-being. In another study, Probst et al. [78] showed that psychological capital attenuated the adverse impact of job insecurity on employee task performance. Other studies have conceptualized psychological capital as a predictor, rather than a moderator, which helps to mitigate the negative effect of job insecurity. For example, Chiesa et al. [30] investigated a sample of young Italians workers and found that these workers’ psychological capital can mitigate perceptions of job insecurity through perceptions of employability. This is consistent with a recent meta-analysis by Jiang et al. [34], which found that psychological capital, among other psychological resources, can significantly alters one’s perceptions of job insecurity. While a recent experimental study showed that a psychological capital intervention (PCI) can significantly decrease job insecurity and stress levels among telecom employees in India [32], research has yet to examine whether perceived job insecurity may mediate the relationship between psychological capital and job stress. This leads to the following hypothesis.

**Hypothesis** **3.**
*Perceived job insecurity mediates the relationship between psychological capital and job stress at the individual level of analysis.*


Apart from the individual-level investigation of psychological capital, several previous studies have indicated that the virtuous influence of psychological capital may also emerge at the work-unit level [35,79,80]. In particular, “collective” psychological capital can be viewed as “a synergy of the interaction and dynamic coordination between group members that comprises collective efficacy, collective optimism, collective hope, and collective resilience” [80]. In their qualitative study, Lansisalmi et al. [14] refer to this as collective coping—the learned, uniform responses that members within the same work unit use to remove the stressor, to alter the interpretation of the stressful situation, or to alleviate the shared negative feelings that result from the prospects of losing jobs. For example, employees may tell stories of the “good old days” to cope with the looming risk of unemployment. It has been shown that authentic leadership [79] and shared leadership [80] can influence collective psychological capital, which in turn can influence several positive outcomes. To date, no research has examined whether collective psychological capital may also exert its virtuous influence by mitigating employees’ shared perceptions of job insecurity and job stress. Thus, we formulate the following hypothesis.

**Hypothesis** **4.**
*Perceptions of job insecurity climate mediate the relationship between collective psychological capital and collective stress at the branch level of analysis.*


## 3. Methods

### 3.1. Sample and Procedures

To test the study hypotheses, we collected survey data from 10 major banks in the southern region of Thailand. Prior to the outbreak of COVID-19, these banks had already begun downsizing their physical operations and workforce while the advent of COVID-19 radically accelerated these changes. It was estimated that at least 25% of bank employees would be impacted by organizational downsizing during COVID-19 [2]. Survey questionnaires were hand-distributed to 700 bank employees in 70 branches. About 10 employees from each branch were asked to participate in the study but this number may vary for each branch depending on their availability. Nevertheless, it was important for us keep the number of employees per branch higher than 5 in order that a multilevel analysis could be reliably conducted. We also took into account the number of clusters (i.e., branches) that is appropriate for multilevel modeling (at least 50–100 clusters). This procedure was based on a multi-stage sampling design in which bank branches were purposively sampled from several districts in four provinces in the South of Thailand.

The respondents in each branch were asked to respond with information on their psychological capital, perceptions of job insecurity and job stress, as well as their demographic information. Completed questionnaires were returned to the researcher on the same day. Of the 700 questionnaires distributed, a total of 520 usable questionnaires were returned from 53 branches with response rates of 74% and 76%, respectively. The average number of respondents per bank branch was 10, which ranged from 7 to 10. About 76% of respondents were female and 80% of them had received a bachelor’s degree. About 91% of the respondents were permanent workers, whereas the rest were temporary workers. Furthermore, 56.3% of them were “operation” officers and the rest were “senior” workers. Their average age was 33.9 years, ranging from 21 to 59 years, with a standard deviation (*SD*) of about 7.2 years. The average organizational tenure was 7.8 years (ranging from 1 to 40 years), with a *SD* of 6.6 years.

### 3.2. Survey Measures

All the survey instruments used by this study were originally in English. Thus, back translation was conducted [81]. Psychological capital (individual level: α = 0.95; branch level: α = 0.96) was assessed using 12 items from the Psychological Capital Questionnaire with a 6-point Likert scale ranging from 1—strongly disagree to 6—strongly agree [29]. Job insecurity (individual level: α = 0.88; branch level: α = 0.94) was measured with a 4-item job insecurity scale developed by De Witte [82] with a 5-point Likert scale ranging from 1—strongly disagree to 5—strongly agree. Finally, job stress (individual level: α = 0.88; branch level: α = 0.92) was measured using 10 items from the perceived stress scale [83], which was based on a 5-point Likert scale ranging from 1—never to 5—very often. We also controlled for several variables at the individual level of analysis, which have been shown to predict job stress including gender (1 = female, 0 = male), age, education levels, tenure (in years), salary (in Baht), contract types (1 = permanent, 0 = temporary), and hierarchical positions (1 = senior, 0 = operation) [84].

### 3.3. Bivariate Correlations and Descriptive Statistics

Table 1 and Table 2 reported the bivariate correlations, means, and standard deviations (*SDs*) of the study variables. As can be seen in both tables, the bivariate correlations at both individual and branch levels were in the predicted directions, except for the correlation between psychological capital and job insecurity (*r* = −0.15, *p* > 0.05) and that between psychological capital and job stress (*r* = −0.10, *p* > 0.05) at the branch level.

### 3.4. Analytic Procedure

We used multilevel structural equation modeling (MSEM) to test the hypotheses in Mplus Version 7.2 [85]. MSEM can simultaneously generate latent variables at both the between-group and within-group levels [86]. In particular, we examined whether the impacts of psychological capital and job insecurity on job stress may vary across the 53 bank branches while also examining the within-branch effect of psychological capital, job insecurity, and job stress. This multilevel design allows us to examine whether it is the shared perceptions, individual perceptions, or both, that cause job stress.

As discussed below, we first tested whether there was sufficient variance in the variables to be modeled at the branch level. Then the measurement model was examined using multilevel confirmatory factor analysis (MCFA). The indicators to test model fit are chi-square, comparative fit index (CFI), Tucker–Lewis’s index (TLI), root mean square error of approximation (RMSEA), and level-specific information for the standardized root mean square residual (SRMR) index [87,88]. Finally, this study conducted MSEM to test the proposed theoretical model [89].

## 4. Results

### 4.1. Multilevel Considerations

Following LeBreton and Senter [90], we used the intraclass correlation coefficient (ICC(1)) and the within-group agreement measure (rwg(j)) to examine whether the study variables can be modeled at the branch level. In particular, ICC(1) indicates a ratio between within-group and between-group variability, which should be higher than 0.05, whereas the rwg(j) was used to determine the level of inter-rater agreement, which should be at least 0.70 [91]. The results showed that the ICC(1) for job stress was 0.24, whereas the median rwg(j) was 0.92. For psychological capital, the ICC(1) was 0.15, whereas the median rwg(j) was 0.90. Finally, for job insecurity, the ICC(1) was 0.23 and the median rwg(j) was 0.90. Based on these results, we concluded that it was appropriate to model all the theoretical constructs at both the individual and branch levels of analysis.

### 4.2. Measurement Models

To obtain a good model fit, both CFI and TLI values need to be greater than 0.90, whereas RMSEA and SRMR values should be less than 0.08; normed fit chi-square (χ2/df) should be less than 3 [92]. From the results of MCFA, the hypothesized model shows an acceptable fit to data (χ2/(404) = 2.190, *p* = 0; RMSEA = 0.048; CFI = 0.953; TLI = 0.947; SRMR = 0.033 (within-level) and 0.134 (between-level)). This model was significantly better than other alternative models. For example, when psychological capital was combined with job stress, it resulted in a significantly worse model fit (χ2/(584) = 3.5, *p* = 0.00; RMSEA = 0.069; CFI = 0.880; TLI = 0.867; SRMR = 0.111 (within-level) and 0.563 (between-level)). Thus, the three-factor was accepted as the best fitting model.

Convergent validity was assessed by examining factor loading of the measurement items [93]. As can be seen in Table 3, all the factor loadings of the constructs at the individual level were above 0.50, ranging from 0.60 to 0.99. The average variance extracted (AVEs) ranged from 0.57 to 0.93, composite reliabilities (CRs) also ranged from 0.86 to 0.98, which are above the recommended value of 0.60 [94]. At the between-branch level, the factor loadings of the constructs ranged from 0.77 to 1.02. Furthermore, the AVEs ranged from 0.87 to 1.00. The CRs also ranged from 0.96 to 1.00. The square roots of each AVE were greater than the correlations shared between the construct and other constructs in the model. According to Fornell and Larcker [95], the discriminant validity of the constructs in this study can be supported.

### 4.3. Structural Models

The fit of the proposed theoretical model (without the direct path from psychological capital to job stress) was acceptable (χ2/(546) = 1.97, *p* = 0.000, CFI = 0.949, TLI = 0.943; RMSEA = 0.042). A partial mediation model, which considered the direct path was also tested. The overall fit of this alternative model (χ2/(544) = 1.94, *p* = 0.00, CFI = 0.947, TLI = 0.940; RMSEA = 0.043) was not significantly better than the hypothesized model, which was more parsimonious. Thus, the hypothesized structural model was accepted as the best fitting model.

As can be seen from the Table 4 and Figure 1, at the individual level of analysis, psychological capital was negatively related to perceptions of job insecurity (*b* = −0.150, *p* < 0.05), whereas perceptions of job insecurity were also positively related to job stress (*b* = 0.364, *p* < 0.001). This provides support to hypotheses 1 and 3. At the branch level, perceived job insecurity was also positively related to job stress (*b* = 0.745, *p* < 0.001). However, psychological capital was not significantly related to perceptions of job security climate (*b* = −0.200, *p* > 0.10). Thus, hypothesis 2 was supported, whereas hypothesis 4 was not.

We also examined the influence of demographic variables on perceptions of job insecurity and job stress at the individual level of analysis. As can be seen in Table 4, we found that longer-tenured employees perceived lower levels of job insecurity (*b* = −0.165, *p* < 0.05). Furthermore, senior employees were found to perceive lower levels of job insecurity than did operational employees (*b* = −0.143, *p* < 0.05). We also found that the female employees experienced higher levels of job stress than did male employees (*b* = 0.184, *p* < 0.05). Finally, senior employees were found to have lower levels of job stress in comparison with operation employees (*b* = −0.176, *p* < 0.05).

In terms of the indirect effect (Table 5), the results showed that, at the individual level of analysis, the indirect effects of psychological capital on job stress via job insecurity was significant (−0.054; SE = 0.027; 95%; confidence interval (CI) = −0.099, −0.015). However, at the branch level, the indirect effect was non-significant (−0.149; SE = 0.225; 95%; CI = −0.582, 0.262). Overall, the proposed model can explain 11.3 percent of the variance in perceptions of job insecurity and 29.1 percent of the variance in job stress at the individual level of analysis, whereas 66.8 percent of the variance in job stress was explained at the branch level.

## 5. Discussion

Our study investigated whether perceptions of job insecurity can cause job stress among bank tellers in Thailand and whether psychological capital can help to alleviate these negative perceptions. The study hypotheses were fully supported at the individual level of analysis. At the branch level, perceptions of job insecurity climate also emerged to influence job stress. Although we found the presence of collective psychological capital at the branch level, it did not significantly alleviate perceptions of job insecurity climate. Several demographic factors were also found to influence individuals’ perceptions of job insecurity and job stress.

### 5.1. Implications

The COVID-19 pandemic has accelerated the pace of automation in various automatable occupations, such as the banking workforce, putting workers at a greater risk of being permanently automated [53]. In line with previous research, our study findings confirm that perceptions of job insecurity can be consequential for employees’ well-being. As the Transactional Theory of Stress and Coping [28] suggests, the prospect of losing jobs may create feelings of chronic stress due to a prolonged exposure to the perceived threat of job insecurity which can lead to a depletion of psychological resources. Indeed, it is important to acknowledge that, for individuals, job insecurity not only signifies their loss of personal income and employment benefits, but also the loss of social status and social ties outside the family sphere.

However, our work extends previous research by shedding light on the influence of “job insecurity climate” on job stress above and beyond the influence of “individual perceptions of job insecurity”. To date, relatively few studies have examined the impact of job insecurity on job stress at both the individual and work-unit levels of analysis. Our study responds to Shoss’s [96] call for research on job insecurity climate using a multilevel approach. Indeed, while individuals may have personal, private concerns about the possibility of losing jobs, it must be acknowledged that those within the same work units have plenty opportunity to interact, discuss, and share their personal experiences, whether positive or negative, which in turn could lead to a gradual progression toward perceptual homogeneity [97]. As our findings showed, perceptions of job insecurity can be contagious at the branch level of analysis.

These findings clearly suggest that organizations need to be transparent with employees about current organizational changes in order that they have a clear understanding of their career prospects, which will allow them to make personal and professional preparations for the future. Indeed, job insecurity perceptions are largely characterized by feelings of anxiety and uncertainty about future employment, and it is not only up to the employee, but also the employer, to address these important concerns. Indeed, past research indicates that employees are not only concerned about the objective employment outcomes but also about the manner in which organizational decisions are made and implemented [98,99]. As a result, organizations could provide opportunities for employees to engage in constructive information exchange, which could allow them to regain a sense of control over their work situation [36].

As our findings indicate, these efforts could be targeted at lower-level and shorter-tenured employees who tend to experience higher levels of job insecurity. This is consistent with the findings from the meta-analysis by Cheng and Chan [84], which indicates that employees with shorter organizational tenure may be more strongly affected by perceptions of job insecurity than those with longer tenure. Indeed, low-level workers who have short tenure are typically less economically secure and they may have more to lose in comparison to those who are senior and longer-tenured. Interestingly, our findings indicate that female employees tend to experience higher levels of job stress. From the job dependence perspective, it is plausible that men have higher levels of occupational mobility than women and the threat of losing jobs could be less threatening for men [13].

Importantly, our findings indicate that psychological capital can play an instrumental role in helping individuals cope with perceptions of job insecurity, which, in turn, reduces their job stress. In line with Lazarus and Folkman’s [28] Theory of Stress and Coping, our findings indicate that individuals with high psychological capital are less vulnerable to job insecurity. To the extent that individuals’ psychological capital is malleable, organizations should employ strategies to foster this important psychological resource. As shown in previous research [32], psychological capital training can be conducted to alleviate perceptions of job insecurity. In particular, Patnaik et al. [32] showed that a two-hour training session on psychological capital can have a profound impact on employees’ perceived job insecurity even after a three-month period.

### 5.2. Study Limitations

Despite the study contributions, this study is not without its limitations. First, as with most research in this area, the measurements of the variables in this study were based on employees’ self-reports, which may raise concerns about common method bias (CMB). Although we followed several procedures to mitigate this concern [100], future research should collect multiple sources of data and use longitudinal design. Secondly, although our measurements of psychological capital and job insecurity were in line with previous research, which aggregated individual-level perceptions to derive unit-level constructs based on the direct consensus compositional model [101], future studies may consider measuring these constructs using a referent shift (e.g., using a referent “my work unit”) [35]. This could be one of the reasons why psychological capital showed a relatively lower level of variability at the branch level and why it did not exert a significant effect on job insecurity climate. Thirdly, our study did not consider how psychological capital actually influence perceptions of job insecurity. Future research may wish to consider the different coping strategies proposed by Folkman and Lazarus [77]. It is plausible that individuals high on specific components of psychological capital may prefer certain strategies over the others. Furthermore, we only examined job stress as an outcome of job insecurity; however, job insecurity is a powerful job stressor which could generate various negative psychological and physical outcomes [6,7,8,9,10]. Future research may wish to examine other interesting outcomes of job insecurity, such as insomnia. Finally, it is important to acknowledge that those with high psychological capital may be more prone to engage in active job search. It would be thus interesting to examine whether job search preparedness may explain why psychological capital can reduce perceptions of job insecurity.

## 6. Conclusions

Our study contributes to the literature by investigating the influence of job insecurity on job stress as well as the antecedent role of psychological capital. We found significant variation in the levels of psychological capital, job insecurity, and job stress across the bank branches. This is among the very first studies to highlight the important roles of these constructs at both individual and work-unit levels of analysis.

## Figures and Tables

**Figure 1 behavsci-12-00168-f001:**
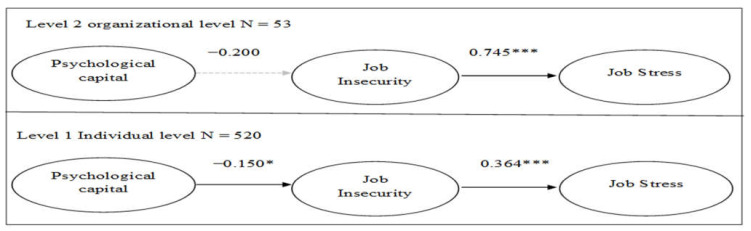
Multilevel structural equation modeling (MSEM) results. Note: * *p* < 0.05. ** *p* < 0.01. *** *p* < 0.001. Dashed lines represent non-significant paths.

**Table 1 behavsci-12-00168-t001:** Means, standard deviations, and correlations (individual level: *N* = 520).

Variables	M	*SD*	1	2	3	4	5	6	7	8	9
1. Psychology Capital	4.68	0.95	(0.95)								
2. Job Insecurity	2.77	1.06	−0.15 **	(0.88)							
3. Job Stress	2.75	0.89	−0.05	0.45 **	(0.88)						
4. Gender	-	-	0.04	−0.02	0.04	-					
5. Age	33.90	7.23	0.15 **	−0.18 **	0.15 **	−0.13 **					
6. Education	-	-	0.04	−0.01	−0.05	−0.04	0.27 **				
7. Tenure (years)	7.79	6.57	0.13 **	−0.21 **	0.17 **	−0.05	0.80 **	0.24 **			
8. Salary (Baht)	-	-	0.08	−0.11 *	−0.13 **	0.01	0.12 **	0.07	0.08 **		
9. Contract Types	-	-	−0.06	0.01	0.04	−0.06	−0.24 **	−0.15 **	0.20	−0.02	
10. Hierarchical Positions	-	-	−0.01	−0.19 **	−0.23 **	0.03	0.51 **	0.21 **	0.49 **	0.19 **	−0.21 **

Note: * *p* < 0.05. ** *p* < 0.01; for gender, 0 = male, 1 = female workers; for contract types, 0 = permanent workers; 1 = temporary workers; for hierarchical positions, 0 = senior workers, 1 = operation workers. Values in parentheses are Cronbach’s alphas.

**Table 2 behavsci-12-00168-t002:** Means, standard deviations, and correlations (branch level: *N* = 53).

Variables	M	*SD*	1	2	3
1. Collective Psychological Capital	4.68	0.50	(0.96)		
2. Job insecurity Climate	2.77	0.62	−0.15	(0.94)	
3. Collective Job Stress	2.75	0.51	−0.10	0.70 **	(0.92)

Note: * *p* < 0.05. ** *p* < 0.01. Values in parentheses are Cronbach’s alphas.

**Table 3 behavsci-12-00168-t003:** Standardized factor loading scores for multilevel confirmatory factor analysis.

Latent Constructs and Manifest Indicators	Loadings within Level *N* = 520	Loadings between Level *N* = 53
**Psychological capital**	(0.93; 0.98)	(1.00; 1.00)
1. Self-efficacy	0.929	1.007
1.1 I feel confident in representing my work area in meetings with management.	0.801	0.880
1.2 I feel confident contributing to discussions about the organization’s strategy.	0.806	0.984
1.3 I feel confident presenting information to a group of colleagues	0.874	0.971
2. Hope	0.988	1.000
2.1 If I should find myself in a jam at work, I could think of many ways to get out of it	0.846	0.989
2.2 Right now, I see myself as being successful at work	0.805	0.988
2.3 I can think of many ways to reach my current work goals.	0.879	1.000
2.4 At this time, I am meeting the work goals that I have set for myself	0.836	0.981
3. Resilience	0.994	0.983
3.1 I can be “on my own,” so to speak, at work if I have to.	0.861	1.004
3.2 I can get through difficult times at work because I’ve experienced difficulty before	0.837	0.988
4. Optimism	0.943	1.021
4.1 I always look on the bright side of things regarding my job.	0.847	0.956
4.2 I am optimistic about what will happen to me in the future as it pertains to work.	0.892	0.960
**Job Insecurity**	(0.62; 0.86)	(0.87; 0.96)
1. Chances are, I will soon lose my job.	0.798	0.945
2. I am not sure whether I can keep my job.	0.605	0.779
3. I feel insecure about the future of my job.	0.864	1.011
4. I think I might lose my job in the near future.	0.850	0.982
**Job Stress**	(0.57; 0.90)	(0.93; 0.99)
1. In the last month, how often have you been upset because of something that happened unexpectedly?	0.749	0.974
2. In the last month, how often have you felt that you were unable to control the important things in your life?	0.758	0.941
3. In the last month, how often have you felt nervous and “stressed”?	0.788	0.961
4. In the last month, how often have you found that you could not cope with all the things that you had to do?	0.722	0.913
5. In the last month, how often have you been able to control irritations in your life? (Reverse-coded)	0.680	0.994
6. In the last month, how often have you been angered because of things that were outside of your control?	0.821	1.002
7. In the last month, how often have you felt difficulties were piling up so high that you could not overcome them?	0.769	0.980

Note: All loadings are significant at *p* < 0.01. Values in the parentheses are AVE = average variance extracted and CR = composite reliability.

**Table 4 behavsci-12-00168-t004:** Multilevel structural equation modeling (MSEM) analyses.

Estimated Paths	Within Level (*N* = 520)		Between Level (*N* = 53)	
	*b*	*b*	*b*	*b*
	Job Insecurity	Job Stress	Job Insecurity	Job Stress
Main Analyses				
Psychological Capital	−0.150 *	-	−0.200	-
Job insecurity	-	0.364 ***	-	0.745 ***
Control Variables				
Gender (1 = female)	−0.087	0.184 *	-	-
Age	−0.042	−0.003	-	-
Education	0.020	−0.077	-	-
Tenure	−0.165 *	0.003	-	-
Salary	−0.047	−0.001	-	-
Contract types (1 = permanent)	−0.005	−0.252	-	-
Positions (1 = senior)	−0.143 *	−0.167 *	-	-
Explained variance (*R*^2^)				
Job insecurity	0.113 **	-	0.037	-
Job Stress	-	0.291 ***	-	0.668 **

Note: *b* = unstandardized coefficients; * *p* < 0.05. ** *p* < 0.01. *** *p* < 0.001.

**Table 5 behavsci-12-00168-t005:** Indirect effect results.

Mediated Paths	Effect	SE	95% CI	
			LLCL	ULCL
Within level (*N* = 520)				
Psychological Capital → Job insecurity → Job stress	−0.054 *	0.027	−0.099	−0.015
Between level (*N* = 53)				
Psychological Capital → Job insecurity → Job stress	−0.149	0.225	−0.582	0.262

Note: CI refers to confidence interval; LLCI = lower limit confidence interval; ULCI = upper limit confidence interval; to indicate significant indirect effects, the interval of CI should exclude zero; * = significant indirect paths (95% CI). * *p* < 0.05.

## Data Availability

The dataset of this study is available from the corresponding author on reasonable request.

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
