# Peer review of "Fear of Losing Jobs during COVID-19: Can Psychological Capital Alleviate Job Insecurity and Job Stress?"

_behavsci, 2022, doi:10.3390/bs12060168_

Round 1
Reviewer 1 Report
The article is well written, with an extensive theoretical background and sound statistical procedures. The analyses are conducted properly and the data support the conclusions. The only thing which could be improved is an explanation of the relatively limited number of questionnaires collected per branch. The authors indicate ~10 questionnaires sent per branch, and then write about 8-12 questionnaires received from each branch. This is slightly problematic but does not undermine the total merit of the article.
Reviewer 2 Report
The paper is written in a clear and fluent way and the analysis conducted appear to be thorough and rigorous. The methodological approach is well described and suitable for the task at hand. The research aim of the study has been achieved. The results of the analysis are well presented. The identified limitations for the conducted research inquiry were shown. The style of the paper is clear and comprehensible. In short, important good research work. I congratulate the authors.
Some minor comments:
I invite the authors to supplement Introduction with the information about the significance of the study. Thus, in the Introduction, you should say what kind of use you envisage for the study results.
Some more further research avenues could be pointed out in Conclusion.
Reviewer 3 Report
Are the results of the study valid?
To answer this preliminary question, we have asked ourselves two questions:
Regarding the first question, I must say that the main objective of the study is clearly and precisely defined; to test the mediation effect of psychological capital, both individual and collective, between the perception of job insecurity and stress in bank employees, at a time of reorganization of the banking sector.
The fundamental question of the study is well posed, as the reader can easily grasp which population is the object of the study, which risk factors are subject to assessment and the expected results. On the other hand, the question addressed is relevant at the theoretical level; that is, it is accurate from the theoretical framework of “collective psychological capital”. It also has practical implications of interest for the psychological counseling of individuals and groups exposed to transformations in the economic sectors in which they work.
Regarding the second question, it should be noted that the description of the sample recruitment procedure could be improved, as well as the method of distribution and collection of questionnaires.
To this end, the authors could indicate the percentage of employees of the selected entities affected by the employment regulation before their study, and the coincidence of their profile with that of the study sample.
On the other hand, it would be useful to know how exactly the questionnaires were distributed and collected; directly, face-to-face, or indirectly via e-mail or other online means? Such information is useful to assess the response rate obtained and to infer the degree of motivation and commitment of the respondents to the results of the study.
Other issues related to the evaluation of the internal validity of the study are the measurement of the variables and the precision of the results. What we can say in this respect is that both procedures are adequate and consistent with the hypotheses put forward. In addition, they consider possible confounding factors, such as age, sex, salary, etc.
The collective psychological capital can probably be measured in another way that is more consistent with its conception. It is even possible that the variables studied could be measured more objectively. Moreover, that a longitudinal design would be better suited to process variables such as job stress. However, we consider that these aspects do not detract from the work to the point of being an obstacle to the dissemination of the results obtained. For these, with their limitations, are accurate and valuable, both at the theoretical and practical levels. At the theoretical level because they contribute to the consolidation of a little-researched construct of great interest for the understanding of certain group or collective behaviors. At the practical level because the confirmation of the mediation hypothesis of psychological capital represents an opportunity to provide new solutions to problems related to health and quality of working life.
